**Data Availability Statement:** The work was performed under a business arrangement between HealthInfoNet (http://www.hinfonet.org), the

# Identification of patients at risk of new onset heart failure: Utilizing a large statewide health information exchange to train and validate a risk prediction model

**Son Q. Duong**[1]*, **Le Zheng**[1,2], **Minjie Xia**[3], **Bo Jin**[3], **Modi Liu**[3], **Zhen Li**[4,5], **Shiying Hao**[1,2], **Shaun T. Alfreds**[6], **Karl G. Sylvester**[7], **Eric Widen**[3], **Jeffery J. Teuteberg**[8], **Doff B. McElhinney**[1,2], **Xuefeng B. Ling**[2,7]*

1 Clinical and Translational Research Program, Betty Irene Moore Children's Heart Center, Lucile Packard Children's Hospital, Palo Alto, California, United States of America, 2 Department of Cardiothoracic Surgery, Stanford University School of Medicine, Stanford, California, United States of America, 3 HBI Solutions Inc., Palo Alto, California, United States of America, 4 Binhai Industrial Technology Research Institute, Zhejiang University, Tianjin, China, 5 School of Electrical Engineering, Southeast University, Nanjing, Jiangsu, China, 6 HealthInfoNet, Portland, Maine, United States of America, 7 Department of Surgery, Stanford University School of Medicine, Stanford, California, United States of America, 8 Division of Cardiovascular Medicine, Stanford University School of Medicine, Stanford, California, United States of America

* sqduong@gmail.com (SQD); bxling@stanford.edu (XBL)

## Abstract

### Background

New-onset heart failure (HF) is associated with poor prognosis and high healthcare utilization. Early identification of patients at increased risk incident-HF may allow for focused allocation of preventative care resources. Health information exchange (HIE) data span the entire spectrum of clinical care, but there are no HIE-based clinical decision support tools for diagnosis of incident-HF. We applied machine-learning methods to model the one-year risk of incident-HF from the Maine statewide-HIE.

### Methods and results

We included subjects aged $\geq$ 40 years without prior HF ICD9/10 codes during a three-year period from 2015 to 2018, and incident-HF defined as assignment of two outpatient or one inpatient code in a year. A tree-boosting algorithm was used to model the probability of incident-HF in year two from data collected in year one, and then validated in year three. 5,668 of 521,347 patients (1.09%) developed incident-HF in the validation cohort. In the validation cohort, the model c-statistic was 0.824 and at a clinically predetermined risk threshold, 10% of patients identified by the model developed incident-HF and 29% of all incident-HF cases in the state of Maine were identified.

### Conclusions

Utilizing machine learning modeling techniques on passively collected clinical HIE data, we developed and validated an incident-HF prediction tool that performs on par with other

operators of the Maine Health Information Exchange and HBI Solutions, Inc. (HBI) located in California. HIN is a steward of the data on behalf of its members which includes health systems, hospitals, medical groups and federally qualified health centers. The data is owned by the HIN members, not HIN. HIN is responsible for security and access to its members' data and has established data service agreements (DSAs) restricting unnecessary exposure of information. HIN and its board (comprised from a cross section of its members) authorized the use of the de-identified data for this research, as the published research helps promote the value of the HIE and value to Maine residents. The research was conducted on HIN technology infrastructure, and the researchers accessed the de-identified data via secure remote methods. All data analysis and modeling for this manuscript was performed on HIN servers and data was accessed via secure connections controlled by HIN. Access to the data used in the study requires secure connection to HIN servers and should be requested directly to HIN. Researchers may contact Shaun T. Alfreds at salfreds@hinfonet.org to request data. Data will be available upon request to all interested researchers. HIN agrees to provide access to the de-identified data on a per request basis to interested researchers. Future researchers will access the data through exact the same process as the authors of the manuscript.

**Funding:** HBI Solutions, Inc. (HBI) is a private commercial company, and several authors are employed by HBI. HBI provided support in the form of salaries for the authors employed by HBI: MX, BJ, ML, and EW. HealthInfoNet also provided a salary for STA. The funders did not have any additional role in the study design, data collection and analysis, decision to publish, or preparation of the manuscript. The specific roles of these authors are articulated in the 'author contributions' section.

**Competing interests:** KGS, EW and XBL are co-founders and equity holders of HBI Solutions, Inc., which is currently developing predictive analytics solutions for healthcare organizations. MX, BJ, ML, and EW are employed by HBI Solutions, Inc. STA is the Executive Director and Chief Executive Officer CEO of HealthInfoNet. This does not alter our adherence to PLOS ONE policies on sharing data and materials. There are no patents, products in development or marketed products associated with this research to declare.

models that require proactively collected clinical data. Our algorithm could be integrated into other HIEs to leverage the EMR resources to provide individuals, systems, and payors with a risk stratification tool to allow for targeted resource allocation to reduce incident-HF disease burden on individuals and health care systems.

## Introduction

The estimated age-adjusted annual incidence of heart failure (HF) is 0.72% in men and 0.47% in women aged 45 or greater, and among 40 year-olds, the estimated lifetime risk of developing HF is 1 in 5 [1,2]. Once diagnosed, HF has a poor prognosis, with one study estimating median survival of 2.3 years and 1.7 years in men and women, respectively, after a first HF hospitalization [3]. HF imposes a large burden on the healthcare system, with at least 20% of hospital admissions in adults >65 years due to HF [4]. There are several validated risk models available to practitioners to predict progression of disease once chronic HF has been diagnosed, such as the Seattle Heart Failure Model [5], but there is a lack of commonly used models to predict onset of heart failure. Given the major role HF plays in the utilization and cost of healthcare, as well as the potential for risk factor modification to delay progression of disease [6], there is a clinical need to develop tools to predict the onset of first diagnosis of HF in order to identify high-risk patients for targeted early interventions and resource allocation. The widespread adoption of the electronic medical record (EMR) and the linking of these records in health information exchanges (HIEs) allows for widespread collection of administrative and clinical data across multiple settings of clinical care, including the clinic, emergency room, hospital, pharmacy, and laboratory settings. These repositories represent a rich source of data with the potential to apply "big-data" machine learning techniques to aid in the risk stratification of individual patients in an automated fashion that may be implemented in the EMR system itself [7]. The objective of this study was to develop and validate a model to predict the individual one-year risk of developing a first-time diagnosis of HF in the adult population by applying machine learning methodology to a large, statewide HIE database that captures 97% of all EMR encounters in the state of Maine [8].

## Methods

### Database and subject selection criteria

This study was approved by the institutional review board of Stanford University. The dataset was derived from the Maine HIE network, which provides real-time point-of-care access for practitioners to records from patients who visited any of the 35 hospitals, 34 federally qualified health centers, and more than 400 ambulatory practices care facilities. The HIE covers nearly 95% of the population of the state of Maine and is managed by the HealthInfoNet organization [9]. The model was designed to predict a patient's 1-year risk of receiving a first-time diagnosis of HF based off of their prior 1-year of EMR data. Three years of data from November 1, 2015 to October 31, 2018 were analyzed.

Training subjects in the discovery cohort were enrolled between November 1, 2015 and October 31, 2016. Discovery cohort subjects' future one year clinical outcomes were tracked. Only patients ≥40 years of age were considered in analysis. Patients with any prior ICD9 and ICD10 code indicative of HF (S1 Table) were excluded. We limited subjects to individuals with at least one recorded encounter before the beginning of the observation period and

excluded patients with missing demographic and income data (see feature selection below). Subjects in the discovery cohort were randomly split into 1/3 training, 1/3 calibration, and 1/3 performance testing groups. The modeling, calibration and performance blind testing processes were used to minimize over-optimism of the test performance characteristics. The model was then tested in a validation cohort consisting of subjects meeting inclusion criteria in the subsequent year from November 1, 2016 and October 31, 2017. Validation cohort subjects' future one year clinical outcomes were followed to report final model performance results.

## Feature standardization, reduction, and selection

We collected the following features from the database for consideration of potential input model predictors: ICD-9 and-10 billing codes, laboratory data in the last 12 months, medications prescribed in the last 12 months, CPT® codes assigned in the last 12 months, and average income of home ZIP code. We used ICD-10 codes throughout the learning process. ICD-9 codes were used only for historic records to identify the past HF events as exclusion criteria. Average income of in the home ZIP code was calculated from 2010 US Census data [10]. We collected all ICD-10 codes that were assigned to each patient during the prediction period, as well as all laboratory data coded in the LOINC system [11]. All outpatient medication prescriptions during the observation period were collected. Finally, all CPT-4® codes, representing billing codes for outpatient procedures, were collected as well. This raw data collection resulted in a massive number of potential coding features which required data reduction techniques to reduce dimensionality. Medications were mapped to medication class using the Established Pharmacologic Class coding system [12]. Laboratory data were provided from the HIE as "abnormal" and "normal" binary categories due data interoperability challenges requiring raw test values to be converted to binary abnormal/normal categorical variables via comparing test result value against the corresponding care providers' test normal reference range. These aggregated data sources provided 43,906 unique potential model features for inclusion. Given the large dimensionality of this dataset, we performed an experiment to determine whether aggregating the 5-digit ICD-CM10-CM codes into the 3-digit code to the left of the decimal (henceforth known as the ICD-10 subheader code) would improve model performance and reduce dimensionality. Finally, we performed a univariate filtering step to eliminate features associated with the outcome with a chi-squared test p-value >0.2. These features were then utilized as candidate features for selection in the XGBoost algorithm. Of note, the algorithm additionally eliminated unimportant features by only considering features with an importance gain greater than 0. The "gain" implies the relative contribution of the corresponding feature to the model calculated by taking each feature's contribution for each tree in the model. A higher value of this metric when compared to another feature implies it is more important for generating a prediction.

## Outcome definition

Development of HF was defined as new assignment of an ICD-10 code for HF (S1 Table) during either 1 inpatient or 2 separate outpatient encounters during the prediction time period. This case definition has been described by others in large EMR studies [13,14] and we observed roughly similar incidence of HF in our cohort compared to that described in the prospectively collected and physician-adjudicated Framingham Heart Study [2] (S2 Table).

## Model construction and tuning

A supervised machine learning and data mining tool used in several biomedical studies, XGBoost [15–17] was applied to develop the prediction model. XGBoost uses a gradient

boosting technique based on a strategy of additive decision trees. In each iteration, a decision tree-based model is trained to predict the prediction errors of the models trained in previous iterations. The decision tree-based model is optimized by an objective function, which consists of a loss function to minimize the error and a regularization term to avoid overfitting. The final prediction result is a sum of the predictions of all the trees. The technical details of this XGBoost procedure were described elsewhere [8]. A hyper-parameter fine tuning process was applied to improve the performance of the system on training set. The hyper-parameters learning rate (eta), maximum depth of a tree, and the number of estimators were tuned using hyper-optimization techniques based on grid search to combine all possible parameters to be optimized to identify the combination that most improved performance. Default regularization parameters were used (L1 = 0, L2 = 1, gamma = 0). In this process, the discovery (training) set is divided into 10-fold for cross validation, and parallel processing based on grid search was used to increase efficiency and minimize the time of parameter tuning. After hyper-parameter fine tuning process, the optimized hyper-parameter combination was found based on best model performance: the learning rate was set to 0.3, the depth of each tree was set to 5 and number of estimators was set to 500. These optimized parameter was applied for XGBoost on the training set to derive final model.

During training, ten-fold cross validation was used. After training, the prediction results were calibrated to the positive predicted value (PPV) to provide a universal standardized risk measurement. We constructed 2 models, 1 utilizing specific ICD-10 codes and the other utilizing ICD subheader codes. We hypothesized that utilization of ICD10 subheader codes would outperform specific ICD codes as it would reduce dimensionality and noise introduced by variations in provider billing practices.

## Model evaluation

Risk scores were expressed as the predicted probability of development of HF, which is equivalent to the positive predictive value (PPV) of the model. The global model performance was evaluated through development of receiver operating curve (ROC) and calculation of area under the curve (AUC) with 95% confidence interval (CI) calculated through bootstrapping methods. We generated observed versus expected calibration curves to examine model performance across all risk scores. For purposes of application of the model to clinical practice, we selected a risk score of ≥0.05 (≥5% probability of development of HF) as the threshold at which patients would be considered high-risk and flagged as "test positive" in the model. At this threshold we calculated the sensitivity, specificity, PPV, and negative predictive value (NPV). For purposes of reproducibility and standardization, the TRIPOD reporting checklist [18] was utilized and available for review in S3 Table.

## Software and hardware

R version 3.5.0 was used with the packages including but not limited to Xtable, XML, Xgboost, whoami, whisker, Xlsx, tidyverse, tidyselect, yaml, xlsxjars. The Windows Server OS 2012 R2 + was used to support computing boxes with CPU 96 vCores, memory 1 TB, 120 GB drive for the OS and 4 TB drive for data mart storage.

## Exploratory model analyses

We explored whether modifying the outcome definition to require only one inpatient or outpatient diagnosis code substantially changed model results. Additionally, we performed additional analysis in which we changed the data dimensionality reduction technique by removing the univariate chi-squared test filtering step. Additionally a Weighted XGBoost method with

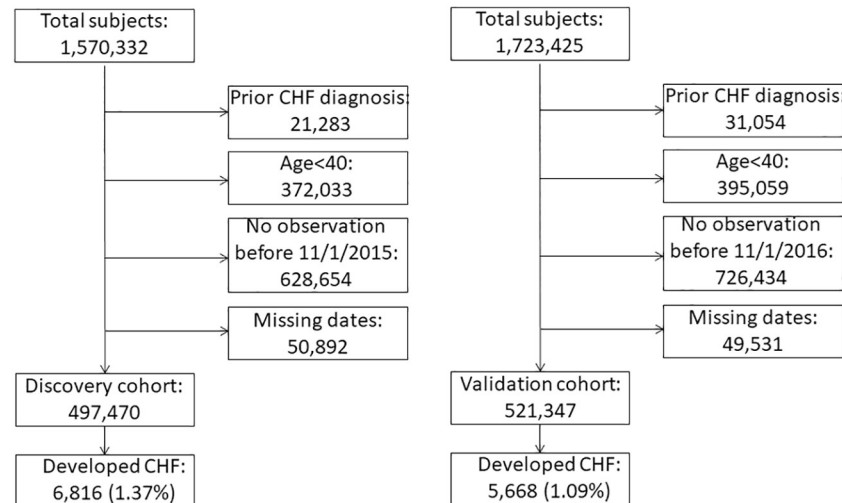

**Fig 1. Cohort identification, discovery and validation cohorts.** Discovery cohort utilized to generate the prediction model, which is subsequently validated on the patient cohort one year after discovery period.

fine tuning of the associated parameters was explored. Since the incidence rate of heart failure is low and the dataset is highly imbalanced, we applied class weighted XGBoost techniques to tune the training algorithm to increase weight to misclassification of the minority class for datasets with a skewed class distribution in order to achieve better performance on heart failure risk prediction problems with a severe class imbalance. In the process, positive class weight hyper-parameter was fine tuned to scale the gradient for the positive class, grid search a range of different class weightings (90, 95, 100, 110, 150) for class-weighted XGBoost and discover the best ROC AUC score. As result, positive class weight was set to 90.

## Results

In the discovery cohort, 497,470 patients met criteria for study inclusion and 521,347 patients met criteria in the validation cohort (Fig 1). The baseline characteristics of the discovery and validation groups are shown in (Table 1). Incident HF was diagnosed in 6,816 (1.37%) individuals in the discovery and 5,668 (1.09%) in the validation cohort. Of the 43,906 possible data features before feature reduction techniques were performed, the algorithm selected 339 for inclusion in the final model (S4 Table). As an example, the top-ranked 25 features included several known to be associated with HF including age; respiratory disorders such as chronic obstructive pulmonary disease; prescriptions for anticoagulation, anti-hypertensive, diuretic, or pulmonary medications; and laboratory markers of abnormal kidney function or glucose homeostasis (Table 2). The use of ICD subheader codes substantially improved model characteristics compared to specific ICD codes, resulting in an increase in the prospective AUC (median, 95% CI) from 0.797 [0.790–0.803] to 0.824 [0.818–0.830]. Thus, ICD subheaders were included for the final model (see Fig 2 for performance characteristics). Model calibration is illustrated in S1 Fig. Calibration from risk scores 0 to 0.05 appeared adequate, but in patients with a risk score >0.05 (5% predicted risk), the model tended to underestimate the risk of developing HF. We therefore used a risk score of 0.05 and higher as a relevant test threshold to classify patients into a higher risk

**Table 1. Baseline characteristics of discovery and validation cohorts.**

| | Discovery cohort | | Validation cohort | |
|---|---|---|---|---|
| | **Heart Failure Group** | **Non-Heart Failure Group** | **Heart Failure** | **Non-Heart Failure Group** |
| | **N = 6,816** | **N = 490,654** | **N = 5,668** | **N = 515,679** |
| Age, mean (SD) | 76 (10.49) | 65 (13.42) | 78 (11.16) | 63 (13.05) |
| Gender, N(%) | | | | |
| Male | 3,953 | 214,775 | 2,864 | 278,742 |
| Female | 2,863 | 275,879 | 2,804 | 236,937 |
| Type 2 diabetes: | | | | |
| Yes | 1,022 | 50,722 | 893 | 59,643 |
| No | 5,794 | 439,932 | 4,775 | 456,036 |
| Essential Hypertension | | | | |
| Yes | 843 | 110,547 | 693 | 134,278 |
| No | 5,973 | 380,107 | 4,975 | 381,401 |
| Chronic kidney disease (CKD) | | | | |
| Yes | 750 | 36,789 | 653 | 45,381 |
| No | 6,066 | 453,865 | 5,015 | 470,298 |

category. This yielded a test sensitivity of 29.2% [95% CI 28.1–30.4%]), specificity of 97.1% [95% CI 97.1–97.2%], positive predictive value of 10.0% [95% CI 9.7–10.4%], and negative predictive value of 99.2% [95% CI 99.1–99.2%]. The relative risk of development of HF in the test positive group was 9.17 times greater than the baseline incidence of 1.09%.

## Exploratory analyses

An additional analysis was performed in which the outcome definition was modified to require only 1 outpatient encounter with coding for HF, and found minor increases in sensitivity, PPV, and AUC (S2 Fig). Another exploratory analysis was performed in which the univariate filtering step was eliminated and further model fine-tuning was added by incorporation of class weight methods. This model selected 234 features (of the original 43,906) as important and showed modest statistically significant improvements in the AUC compared to the original model (validation cohort AUC 0.858 vs. 0.824; p = 0.01). Sensitivity at the clinical threshold was increased, however the specificity was decreased, and overall PPV and NPV were unchanged (S3 Fig).

## Discussion

In this study, we used data from a large, state-wide EMR clinical information exchange with aggregated demographic, medication, laboratory, medical procedure, and socioeconomic data to develop a model to predict the 1-year risk of developing HF in adults ≥40 years of age. The validated model exhibits good discrimination ability (AUC = 0.824), and by incorporating a clinically relevant threshold of a predicted 5% risk or greater, we were able to capture 29% of incident HF cases in the state of Maine from 2017–2018 with a PPV of 10%. This algorithm identifies a population with an over ninefold greater risk of developing HF compared to the baseline population. Put another way, approximately one in ten patients that test positive in this model are predicted to go on to develop heart failure in the next year, compared to one in 100 at baseline. A strength of this model is that it was built upon extant information in the EMR and was designed to be immediately applicable to the EMR as an "early warning" tool for clinicians and patients. It is possible that such a tool could focus practitioners to screen for

**Table 2. Top 25 most important features from final model (of 339 total features selected).**

| Importance Rank | Feature | Feature Class |
|---|---|---|
| 1 | Loop diuretic medication prescribed | Medication |
| 2 | Beta-Adrenergic Blocker prescribed | Medication |
| 3 | Age Group (> = 85) | Demographics |
| 4 | Age Group (75–84) | Demographics |
| 5 | Long term (current) drug therapy | ICD10 Subheader |
| 6 | Other chronic obstructive pulmonary disease | ICD10 Subheader |
| 7 | Age Group (35–49) | Demographics |
| 8 | Age Group (50–64) | Demographics |
| 9 | Essential (primary) hypertension | ICD10 Subheader |
| 10 | Presence of cardiac and vascular implants and grafts | ICD10 Subheader |
| 11 | Age Group (65–74) | Demographics |
| 12 | Vitamin K Antagonist prescribed | Medication |
| 13 | Abnormalities of breathing | ICD10 Subheader |
| 14 | Beta2-Adrenergic Agonist prescribed | Medication |
| 15 | Patient had abnormal blood glucose laboratory test | Laboratory |
| 16 | Hypertensive chronic kidney disease | ICD10 Subheader |
| 17 | Male | Demographics |
| 18 | Encounter for screening for malignant neoplasms | ICD10 Subheader |
| 19 | Angiotensin Converting Enzyme Inhibitor prescribed | Medication |
| 20 | Abnormal Blood Urea Nitrogen laboratory test | Laboratory |
| 21 | Encounter for general exam without complaint | ICD10 Subheader |
| 22 | Patient's Zip Code area has a very low median Income | Demographics |
| 23 | HMG-CoA Reductase Inhibitor prescribed | Medication |
| 24 | Other peripheral vascular diseases | ICD10 Subheader |
| 25 | Abnormal serum creatinine laboratory test | Laboratory |

asymptomatic left ventricular dysfunction, or control modifiable risk factors for HF such as hypertension [19] and lipid disorders [6] which may reduce progression of disease.

Our model is unique in that it only requires a prior years' worth of patient data and utilizes information passively collected within the EMR from several sources to include demographic, pharmacy, inpatient, and outpatient encounters. This is unlike traditional risk prediction tools that rely on historical or laboratory markers collected actively. The boosting algorithm by nature is able to incorporate more features than other reported models derived from typical regression methods, and a total of 339 predictors were utilized as classifiers in our study. The model identified many of the traditionally described epidemiological risk factors for HF including age, hypertension, diabetes, chronic obstructive pulmonary disease, arrhythmia, atherosclerosis, cerebrovascular disease, kidney disease, and obesity. Interestingly, the model also agnostically identified more recently identified novel associations reported in the HF literature such as abnormal iron [20] and vitamin D levels [21]. This demonstrates how machine-learning derived tools applied to large clinical data sets can detect subtle associations, though further exploration is required to study if these are truly causative or contributing factors.

Systematic reviews of clinical HF prediction models [22,23] reported AUC values ranging from 0.71–0.92. These studies utilized logistic regression methods for prediction and all of them relied on actively measured risk factors. There was wide variation in outcome definition, ranging from ICD coding to the "gold standard" Framingham criteria. In the machine-learning literature, the reported AUC values for prediction of HF are comparable or to our findings

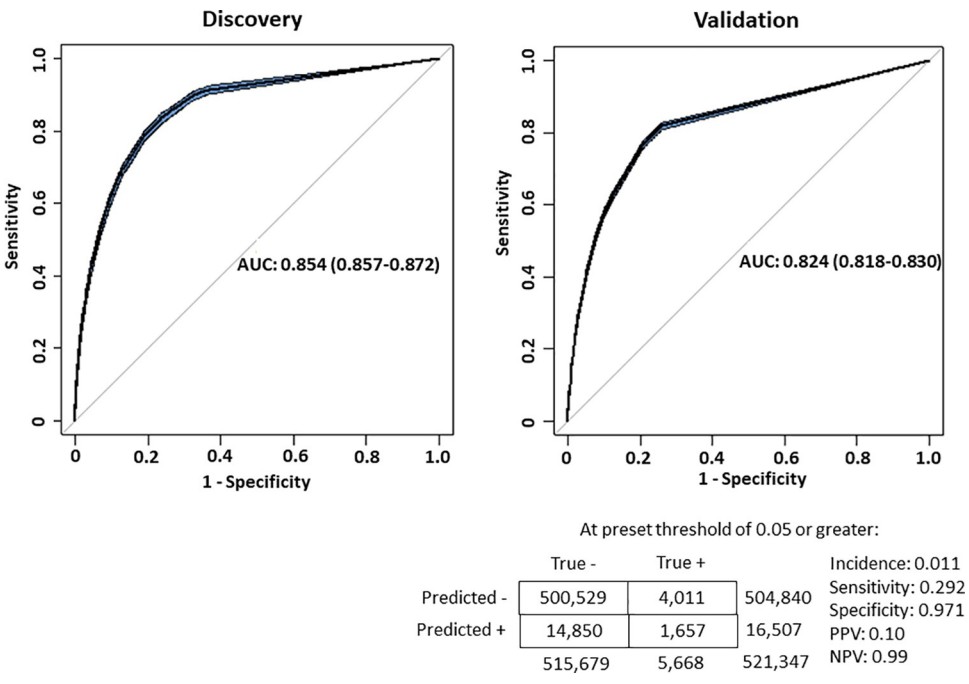

At preset threshold of 0.05 or greater:

|  | True - | True + |  | Incidence: 0.011 |
|---|---|---|---|---|
| Predicted - | 500,529 | 4,011 | 504,840 | Sensitivity: 0.292 |
| Predicted + | 14,850 | 1,657 | 16,507 | Specificity: 0.971 |
|  | 515,679 | 5,668 | 521,347 | PPV: 0.10 |
|  |  |  |  | NPV: 0.99 |

**Fig 2. Final model (discovery and validation) characteristics.** Model test characteristics from the discovery and validation cohorts. Blue shaded area represents 95% Confidence interval. Test characteristics shown at a clinically preset threshold risk score of 0.05 or greater with resultant test characteristics.

[14,24–29]. Direct comparison of models is limited due to differences in study design; namely, the prior studies all used a case-control design, whereas ours was validated on a validation cohort of "all-comers" meeting inclusion criteria, making our model more clinically applicable. Wu et al. [29] trained a model to predict incident HF over a 3-year period from case and control cohorts in large multi-site outpatient group in Pennsylvania. Using a boosting algorithm similar to ours, they reported a median model AUC of 0.78, but did not present a validation cohort. Ng et al. [26] used a matched case-control population of primary care clinics across a large practice in central and northeastern Pennsylvania to model incident HF using random forest modeling. They performed similar feature aggregation techniques to reduce data sparsity, but they used US Center for Medicare & Medicaid Services-derived hierarchical condition categories which aggregate diagnoses into a much more general categories than ICD10 subheaders. They reported an AUC of 0.78 and did not perform validation. Choi et al. [24] and Rasmy et al. [27] both applied a Recurrent Neural Network (RNN) deep learning algorithm to a case-control cohort within large, multi-hospital EMR systems and achieved an AUC of 0.77–0.79 in a validation test subset. Interestingly, Rasmy et al reported that the use of a different clinical classification hierarchical ontology of ICD codes from the US Agency for Healthcare Research and Quality Clinical Classification Software (CCS), was inferior to ICD codes, whereas Choi reported that the use of grouped codes, including CCS, improved prediction. Our results suggest that the ICD10 subheader organizational system yields better prediction than specific ICD codes, which is consistent with our hypothesis that utilizing this system can provide both feature and noise reduction due to variability in provider coding practices. To our knowledge this has not been reported previously in the HF risk prediction literature. Wang et al. [28] reported on models of HF diagnosis using gradient boosting on a matched group of cases and control patients in an outpatient EMR system. Over a shorter prediction window of 180 days and utilizing only ICD9 codes and medications, they achieved an AUC of

0.71. Of note, they reported that the use of principal component analysis to aggregate meaningful input features worked well for small training sets but as the training set size increased the use of aggregation hurt prediction performance whereas we found aggregation using ICD subheader codes to improve performance.

## Potential limitations

As these are real-world data there are continued opportunities to improve model development. We show in additional exploratory analyses that the model has variable performance characteristics depending on choice of outcome definition and data reduction techniques. Removal of the univariate prefiltering step and the addition of methods to deal with class imbalance results in a small increase in the model AUC, but at the expense of model specificity. Decisions on which model is "best" ultimately depends on the clinical needs of the practitioner or health system utilizing the tool. These exploratory analyses show, though, that iterative optimization of our model is important for continued application on real-world data.

The use of ICD coding is both a strength and a weakness of this modeling approach—it allows for effortless computer-driven data collection, but is subject to incomplete data and inaccurate diagnosis. Importantly, the AUC estimates in this study are comparable to other studies that used ICD coding for case definition [13,30–34]. As with all studies based on clinical and administrative information, coding can be incomplete or inaccurate. This may manifest in variations in provider coding of similarly related conditions, incorrect diagnosis, and implicit treatment for HF without explicit coding for HF. Physician coding practices may be variable due to a litany of factors including: incomplete or nonspecific clinical documentation, lack of commonality between coding terminology and disease processes, and discrepancies between coders and health care providers performing other forms of clinical documentation. The effect of this variation on individual risk assessments is difficult to predict, and is likely variable across different disease processes. We attempted to address variation in provider coding practices by aggregating 5-digit ICD-10 CM codes into 3-digit ICD subheader codes, and found greater capture of relevant features. Although domain knowledge manual curation-based feature engineering like ICD10 code hierarchical grouping can increase feature density, it may miss true risk factors at a lower level of the hierarchical structure. In the future, deep learning neural networks-based dimensionality reduction such as autoencoder methods [35,36] for unsupervised training for dimensionality reduction and feature discovery may improve model prediction performance. To address the potential for bias due to incorrect diagnosis of HF, we incorporated a more stringent case definition criteria of 2 outpatient or 1 inpatient criteria, which has been validated by other groups [13,14], and notably in additional exploratory analysis we did not see a significant change in model performance with a less stringent case definition. Additionally, our observed incidence of ~1% in adults aged over 40 years reasonably approximates that described by the gold-standard, physician-adjudicated prospective Framingham Heart Study [2] (S2 Table). Furthermore, any subjects that move into the Maine HIE system with a prior diagnosis of HF made outside of the system might be erroneously included as at-risk for HF development. Other limitations included the inability to collect some clinically relevant details due to incomplete coding, such as race/ethnicity, BMI, smoking status, and vital signs. However, some of these data may be captured indirectly in our model with ICD coding or medication prescription. Finally, implicit treatment for HF without explicit coding of the disease was likely present to some degree in our data set. In other words, we cannot assess the impact of patients who were being treated for HF by practitioners without explicitly being labeled as having HF. Presence of these patients may have inflated the apparent performance of the model but not represent clinically meaningful prediction.

Our model was not designed to be applicable to patients <40 years-old. However, given the vanishingly low incidence of HF in this population, we believe our performance characteristics are more conservatively estimated by reducing class imbalance from the high proportion of negative classifications in this group. Our model was also not designed to account for interactions between features, which has the potential to vastly improve the modeling but would exponentially increase the dimensionality of the feature set beyond our computational capacity. Additionally, our model can only generate predictions on patients that access the healthcare system on at least a yearly basis, and will not work for patients who seek care outside of the CCHIE database. Our model is only designed to predict a one-year risk which might limit the clinical window for intervention in patients. Finally, because we did not attempt to perform an adjusted analysis, it was unclear if some of the algorithm-selected features were important in their own right or simply correlated and collinear with other features. As an example, cataract surgery and eye disorders were identified as predictive, but from a clinical perspective this seems to be more likely correlated with age rather than an independent risk factor for HF, but further exploration is required.

## Conclusion and further studies

In conclusion, we report the development of risk prediction tool for development of HF in adults using a large state-wide CCHIE from the state of Maine. As this CCHIE is based on Orion Health's Rhapsody HL7 integration engine and associated stack, which is widely used in the US, we envision that this model could be automatically incorporated into CCHIEs to analyze the vast troves of data present in the modern EMR and identify, without any active provider intervention, a set of patients >40 years of age that are at substantially higher risk for development of HF than the general population. We envision that this system could be built directly into the EMR to allow healthcare providers of all types to adjust their recommendations, with the goal of possibly delaying progression of disease. Other interested parties, such as payors or managed care systems, may use this tool for targeted resource allocation, and even patients themselves could use such a tool to monitor their own disease risk profiles to encourage lifestyle modification. Before deployment in clinical practice, this model will need to be validated and refined in other large datasets and patient populations. Further work is needed to identify the clinical utility and cost effectiveness of screening with this tool.

## Supporting information

**S1 Fig. Model calibration curve.** The observed versus expected model predictions across all risk score assignments. Yellow shaded area represents 95% confidence interval.
(TIF)

**S2 Fig. Exploratory analysis: Modification of outcome criteria to only 1 inpatient or outpatient heart failure code.**
(TIF)

**S3 Fig. Exploratory analysis: Model performance after addition of class weights and elimination of feature prefiltering.**
(TIF)

**S1 Table. ICD codes for heart failure outcome.**
(DOCX)

**S2 Table. Comparison of incidence of HF using study definition (2 outpatient or 1 inpatient ICD code assignment; 2b) versus observed incidences from the Framingham Heart**

Study (2a).
(DOCX)

S3 Table. TRIPOD prediction algorithm checklist.
(XLSX)

S4 Table. All features considered in final model.
(XLSX)

## Author Contributions

**Conceptualization:** Son Q. Duong, Le Zheng, Minjie Xia, Modi Liu, Karl G. Sylvester, Jeffery J. Teuteberg.

**Data curation:** Son Q. Duong, Le Zheng, Modi Liu, Zhen Li, Eric Widen, Xuefeng B. Ling.

**Formal analysis:** Le Zheng, Minjie Xia, Bo Jin, Modi Liu, Zhen Li, Xuefeng B. Ling.

**Investigation:** Son Q. Duong, Shiying Hao, Jeffery J. Teuteberg.

**Methodology:** Shaun T. Alfreds.

**Project administration:** Shaun T. Alfreds.

**Resources:** Shaun T. Alfreds, Xuefeng B. Ling.

**Supervision:** Karl G. Sylvester, Eric Widen, Doff B. McElhinney.

**Writing – original draft:** Son Q. Duong, Shiying Hao.

**Writing – review & editing:** Son Q. Duong, Le Zheng, Shiying Hao, Karl G. Sylvester, Jeffery J. Teuteberg, Doff B. McElhinney, Xuefeng B. Ling.

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
