## [Decision Letter · Decision Letter 0]

26 May 2021

PONE-D-21-10920

A prospectively validated novel risk prediction model for new onset heart failure utilizing a large statewide health information exchange

PLOS ONE

Dear Dr. Duong,

Thank you for submitting your manuscript to PLOS ONE. After careful consideration, we feel that it has merit but does not fully meet PLOS ONE’s publication criteria as it currently stands. Therefore, we invite you to submit a revised version of the manuscript that addresses the points raised during the review process.

Thank you for the opportunity to edit this. It's a good piece of work, and will add to the literature in this rapidly expanding area. Any more detail that could be added for translatability/reproducibility, the better. Similarly, ensuring the manuscript meets a systematic review checklist would increase the assessed quality. All suggestions are optional and intended to add value and potential impact.

We look forward to receiving your revised manuscript.

Kind regards,

Dylan A Mordaunt

Academic Editor

PLOS ONE

Journal Requirements:

5.Thank you for stating the following in the Competing Interests section:

"I have read the journal's policy and the authors of this manuscript have the following competing interests: Dr. Ling, Mr. Widen, and Dr. Sylvester are co-founders and shareholders of HBI Solutions "

We note that one or more of the authors are employed by a commercial company: HBI Solutions

Additional Editor Comments :

- One of the editors has asked for expansion on details of the HIE, as this would be of interest both for reproducibility but also in terms of translation. I understand that the Maine HIE was based on Orion Health's Rhapsody HL7 integration engine and associated stack, which is a very similar stack to both HealthENet in NSW and CalIndex in California.

- I'm aware that previously Maine HIE had operationalized readmissions predictions based on daily extracts from the HIE, returning predictions to case managers. Although this was quality driven, it's worth considering that this was motivated by CMS funding. This track record would be worth citing.

- I can see how this model would be useful and I think the authors should be commended. The features are somewhat telelogical for the purpose, but it may be that in translation the application is different. Just something for consideration.

- This is a good piece of work. I would suggest that if this were to be included in a systematic review our critically analysed, it would be worth the authors undertaking the Tripod ML checklist or similar- https://www.tripod-statement.org/, so as to ensure to increase the impact, quality etc

Reviewers' comments:

Reviewer's Responses to Questions

**Comments to the Author**

1. Is the manuscript technically sound, and do the data support the conclusions?

Reviewer #1: Yes

2. Has the statistical analysis been performed appropriately and rigorously? 

Reviewer #1: Yes

3. Have the authors made all data underlying the findings in their manuscript fully available?

Reviewer #1: No

4. Is the manuscript presented in an intelligible fashion and written in standard English?

Reviewer #1: Yes

5. Review Comments to the Author

Reviewer #1: Please include a more detailed description of the Health Information Exchange (HIE) - is is designed to provide real-time point of care access to practitioners or is is a passive repository for secondary analysis purposes?

I would like to see a sensitivity analysis that ascertains HF based on only a single outpatient HF code (rather than two outpatient codes) - does this improve sensitivity, PPV and AUC?

Perhaps could add to the discussion potential to use newer survival analysis extensions to deep learning models which might require less dimensionality reduction.

Could expand a bit more on how why there is variation in provider billing practices, how this influences ICD coding, and how this might impact the predictive models.

6. PLOS authors have the option to publish the peer review history of their article (what does this mean?). If published, this will include your full peer review and any attached files.

Reviewer #1: **Yes: **Louisa R Jorm

---

## [Author Response · Author response to Decision Letter 0]

18 Jul 2021

Please see the Editor and Reviewer response below

Journal Requirements:

The work was performed under a business arrangement between HealthInfoNet (http://www.hinfonet.org), the operators of the Maine Health Information Exchange and HBI Solutions, Inc. (HBI) located in California. By business arrangement we mean HBI is a contracted vendor to HealthInfoNet (HIN), and HBI is under contract to deploy its proprietary applications and risk models on the HIN data for use by HIN members. HIN is a steward of the data on behalf of its members which includes health systems, hospitals, medical groups and federally qualified health centers. The data is owned by the HIN members, not HIN. HIN is responsible for security and access to its members' data and has established data service agreements (DSAs) restricting unnecessary exposure of information. HIN and its board (comprised from a cross section of its members) authorized the use of the de-identified data for this research, as the published research helps promote the value of the HIE and value to Maine residents. 

HBI receives revenue for providing this service, which is performed remotely. HBI does not own or have access to the data outside of providing services to HIN. HIN manages and controls the data within its technology infrastructure. The research was conducted on HIN technology infrastructure, and the researchers accessed the de-identified data via secure remote methods. All data analysis and modeling for this manuscript was performed on HIN servers and data was accessed via secure connections controlled by HIN. 

Access to the data used in the study requires secure connection to HIN servers and should be requested directly to HIN. Researchers may contact Shaun T. Alfreds at salfreds@hinfonet.org to request data. Data will be available upon request to all interested researchers. HIN agrees to provide access to the de-identified data on a per request basis to interested researchers. Future researchers will access the data through exact the same process as the authors of the manuscript.

 As this is analysis of a third-party dataset, we are unable to provide a dataset for upload (see above)

5.Thank you for stating the following in the Competing Interests section:

"I have read the journal's policy and the authors of this manuscript have the following competing interests: Dr. Ling, Mr. Widen, and Dr. Sylvester are co-founders and shareholders of HBI Solutions "

We note that one or more of the authors are employed by a commercial company: HBI Solutions

- Funding Statements: 

The authors received no specific funding for this work. 

HBI Solutions, Inc. (HBI) is a private commercial company, and several authors are employed by HBI. HBI provided support in the form of salaries for the authors employed by HBI: MX, BJ, ML, and EW. HBI did not have any additional role in the study design, data collection and analysis, decision to publish, or preparation of the manuscript. The specific roles of these authors are articulated in the ‘author contributions’ section.

 Conflict of Interest Disclosures: 

We have the following interests: KGS, EW and XBL are co-founders and equity holders of HBI Solutions, Inc., which is currently developing predictive analytics solutions for healthcare organizations. MX, BJ, ML, and EW are employed by HBI Solutions, Inc.

From the Departments of Surgery, Stanford University School of Medicine, Stanford, California, KGS and XBL conducted this research as part of a personal outside consulting arrangement with HBI Solutions, Inc. The research and research results are not, in any way, associated with Stanford University.

There are no patents, further products in development or marketed products to declare. This does not alter our adherence to all the PLOS ONE policies on sharing data and materials, as detailed online in the guide for authors.

Additional Editor Comments :

- One of the editors has asked for expansion on details of the HIE, as this would be of interest both for reproducibility but also in terms of translation. I understand that the Maine HIE was based on Orion Health's Rhapsody HL7 integration engine and associated stack, which is a very similar stack to both HealthENet in NSW and CalIndex in California.

Thank you for the helpful suggestion. We have incorporated this comment into the discussion section line 251.

“As this CCHIE is based on Orion Health's Rhapsody HL7 integration engine and associated stack, which is widely used in the US, we envision that this model could be automatically incorporated into CCHIEs…”

- I'm aware that previously Maine HIE had operationalized readmissions predictions based on daily extracts from the HIE, returning predictions to case managers. Although this was quality driven, it's worth considering that this was motivated by CMS funding. This track record would be worth citing.

Thank you for the valuable input. We have included this citation in our manuscript on line 38. 

- I can see how this model would be useful and I think the authors should be commended. The features are somewhat telelogical for the purpose, but it may be that in translation the application is different. Just something for consideration.

Thank you, we agree that in translation of the model into clinical practice, some of the proposed applications might change. 

- This is a good piece of work. I would suggest that if this were to be included in a systematic review our critically analysed, it would be worth the authors undertaking the Tripod ML checklist or similar- https://www.tripod-statement.org/, so as to ensure to increase the impact, quality etc

Thank you for the suggestion. Our understanding is that the TRIPOD-ML checklist has not been released yet, however we have included the TRIPOD checklist as a supplemental table and included a reference to the table in the methods section on line 111.

“For purposes of reproducibility and standardization, the TRIPOD reporting checklist was utilized and available for review in Supplemental Table”

 

5. Review Comments to the Author

Reviewer #1: Please include a more detailed description of the Health Information Exchange (HIE) - is designed to provide real-time point of care access to practitioners or is a passive repository for secondary analysis purposes?

- The HIE is designed to provide real-time point of care access to practitioners. This is clarified in the methods section line 45.

“The dataset was derived from the Maine HIE network, which provides real-time point-of-care access for practitioners to records from patients who visited any of the 35 hospitals, 34 federally qualified health centers, and more than 400 ambulatory practices care facilities.”

I would like to see a sensitivity analysis that ascertains HF based on only a single outpatient HF code (rather than two outpatient codes) - does this improve sensitivity, PPV and AUC?

- Thank you for the suggestion. We have performed this analysis and found minor increases in sensitivity, PPV, and AUC, but overall the performance is not significantly changed. Please see addition to results on line 134 and Supplementary Figure 2. 

“A sensitivity analysis was performed in which the outcome definition was modified to require only 1 outpatient encounter with coding for HF, and found minor increases in sensitivity, PPV, and AUC (Supplemental Figure 2).”

Perhaps could add to the discussion potential to use newer survival analysis extensions to deep learning models which might require less dimensionality reduction.

- Thank you for this excellent suggestion. We have added discussion of deep learning techniques for handling high-dimensional data (line 217)

“Although domain knowledge manual curation-based feature engineering like ICD10 code hierarchical grouping can increase feature density, it may miss true risk factors at a lower level of the hierarchical structure. In the future, deep learning neural networks-based dimensionality reduction such as autoencoder methods {Hinton, #82;Kiarashinejad, 2020 #81} for unsupervised training for dimensionality reduction and feature discovery may improve model prediction performance”

Could expand a bit more on how why there is variation in provider billing practices, how this influences ICD coding, and how this might impact the predictive models.

Thank you for the suggestion. We have added the following to the discussion/limitations to address this concern (line 211)

“Physician coding practices may be variable due to a litany of factors including: incomplete or nonspecific clinical documentation, lack of commonality between coding terminology and disease processes, and discrepancies between coders and health care providers performing other forms of clinical documentation. The effect of this variation on individual risk assessments is difficult to predict, and is likely variable across different disease processes.”

---

## [Decision Letter · Decision Letter 1]

23 Aug 2021

PONE-D-21-10920R1

A prospectively validated novel risk prediction model for new onset heart failure utilizing a large statewide health information exchange

PLOS ONE

Dear Dr. Duong,

Thank you for submitting your manuscript to PLOS ONE. After careful consideration, we feel that it has merit but does not fully meet PLOS ONE’s publication criteria as it currently stands. Therefore, we invite you to submit a revised version of the manuscript that addresses the points raised during the review process.

We look forward to receiving your revised manuscript.

Kind regards,

Dylan A Mordaunt

Academic Editor

PLOS ONE

Journal Requirements:

Additional Editor Comments:

Thank you for your submission and amendments. We have received some additional feedback with regards to methods as per the reviewers. Whether these are minor or major suggestions is a matter of perspective, but they are all worth considering. In particular it would be useful to describe how and why this model is different from previous models in the field. All suggestions are addressable. I look forward to receiving your resubmission.

Reviewers' comments:

Reviewer's Responses to Questions

**Comments to the Author**

1. If the authors have adequately addressed your comments raised in a previous round of review and you feel that this manuscript is now acceptable for publication, you may indicate that here to bypass the “Comments to the Author” section, enter your conflict of interest statement in the “Confidential to Editor” section, and submit your "Accept" recommendation.

Reviewer #2: (No Response)

Reviewer #3: (No Response)

Reviewer #4: (No Response)

Reviewer #5: (No Response)

2. Is the manuscript technically sound, and do the data support the conclusions?

Reviewer #2: Partly

Reviewer #3: Partly

Reviewer #4: Partly

Reviewer #5: Partly

3. Has the statistical analysis been performed appropriately and rigorously? 

Reviewer #2: N/A

Reviewer #3: Yes

Reviewer #4: Yes

Reviewer #5: I Don't Know

4. Have the authors made all data underlying the findings in their manuscript fully available?

Reviewer #2: No

Reviewer #3: No

Reviewer #4: Yes

Reviewer #5: No

5. Is the manuscript presented in an intelligible fashion and written in standard English?

Reviewer #2: Yes

Reviewer #3: Yes

Reviewer #4: Yes

Reviewer #5: Yes

6. Review Comments to the Author

Reviewer #2: Some previous comments have been addressed. In my point of view, there are some major points that still need to be addressed to meet the quality for publication:

1. The manuscript itself lacks a lot of literature review on related works.

2. The authors should compare the performance results to previous studies on the same dataset.

3. The authors should propose more feature selection techniques to find out the optimal ones.

4. How did the authors perform hyperparameter optimization of the models?

5. Machine learning-based model (i.e., XGBoost) has been used in previously biomedical studies such as PMID: 31987913 and PMID: 32942564. Thus, the authors are suggested to refer to more works in this description to attract a broader readership.

6. There must have space before reference number.

Reviewer #3: Aim:

This paper developed a risk prediction tool to detect incident heart failure in adults using a large state-wide CCHIE from the state of Maine, USA.

A tree-boosting algorithm was trained in order to model the probability of incident heart failure in year two from data collected in year one, and then prospectively validated in year three.

This paper tackles a very important problem that could be solved by using existing routinely collected data. In addition, it shows how difficult could be for an algorithm to predict HF based on administrative and billing codes such ICD10. I enjoyed reading the paper.

The model obtains a high specificity but a very low sensitivity. This could be expected as the classes are highly imbalanced, making this problem a difficult one. If the algorithm classifies everyone as healthy, it will have already very high specificity. I would invite the authors to tackle this problem by using some of the machine learning techniques that deal with imbalanced datasets, for example, using a Class Weighted XGBoost or Cost-Sensitive XGBoost.

In addition, given the vast amount of data (labs, procedures, medicines,...) the authors could use other markers in addition to ICD10 to better define HF. This will include some work with clinicians.

Finally, the authors have seemed to just "plug" the XGBoost to the data without tuning any hyper-parameter. I would also invite the authors to re-visit this and tune some of the most important hyper-parameters of the XGBoost. This could significally improve the performance of the predictive algorithm.

Please find some other suggestions below:

Abstract:

1. Methods and Results: "A tree-boosting algorithm was developed": A tree-boosting algorithm was trained, rather than developed.

2. Conclusions: See my comment below regarding the term "prospectively validated".

Methods

Database and subject selection criteria:

1. I would use the terms "training and test" sets for the sets that you use for training and testing. I am not 100% clear which part of the data you use for training and which one for testing. Explain this in detail with dates and number of records. Table 1 will be ideal for that.

Every machine learning algorithm is tested in some data that the model hasn't seen during the training, but I wouldn't call it "prospectively validated" unless you collect the data prospectively, which as far as I understood, it is not the case here. This paper may be of interest: https://jamanetwork.com/journals/jamanetworkopen/fullarticle/2760438 (Prospective and External Evaluation of a Machine Learning Model to Predict In-Hospital Mortality of Adults at Time of Admission

Nathan Brajer). They used the terms training and test sets to build the model, and then, they prospectively validated by validation the model with real-time data: "The model was integrated into the production electronic health record system and prospectively validated on a cohort of 5273 hospitalizations representing 4525 unique adult patients admitted to hospital A between February 14, 2019, and April 15, 2019." Therefore, in my opinion, you use the data as if it was collected prospectively, which is exactly the idea behind any test set, but you didn't integrate the model into production and validate prospectively.

I would remove the term "prospectively validated" from the manuscript and replace it with "test the algorithm" or validate it in the test set.

2. "Subjects in the model building/training cohort were randomly split into a 2/3 training and 1/3 retrospective prediction group, which was used to train separate models under differing feature sets (see below)." I was confused with the definition of the training and test set. In addition ,Table 1 doesn't specify the dates or the amount of patients (2/3 and 1/3) that falls in each group.

Table 1 says that the training cohort contains 497,470 patients, but Figure 1 says the 497,470 were divided into "observation period" (usually this is the training set) and prediction period (this is NOT usually part of the training set).

Finally, the term "validation cohort" is a bit confusing too, since "validation or development set" is commonly used for hyperparameter tuning. As stated above, I would change it to training and test. If validation cohort is preferred, please state clearly, with dates and number of patients, what sets you used for training, test and external testing or "validation". But I don't think it is either external (since the data comes from the same system) or prospective (as it is not collected prospectively).

Feature selection and preprocessing:

1. Did you convert the ICD9 to ICD10 codes? You talked about using 5 or 3 digits with the ICD10. How do you deal with the ICD9 codes?

2. "Finally, we performed a univariate filtering step to eliminate features associated

78 with the outcome with a chi-squared test p-value >0.2." You could use XGBoost for both purposes, 1) feature/dimensionality reduction (https://machinelearningmastery.com/feature-importance-and-feature-selection-with-xgboost-in-python/) and 2) Final model. It would be interesting to see which features the XGBoost presents as the most relevant.

3. Did not you include patient characteristics such as gender? I think males tend to have higher incidence of HF.

4. "Medications were mapped to medication class using the Established Pharmacologic Class coding system". Were medications treated as binary 1-Medication was taken, 0 -Wasn't used, or did you use the amount of units, tablets,...?

In supplementary Table 3, the medications contain this text: Patient had *** medications , which makes me think you consider a continuous number. Please clarify.

5. Exactly the same for lab test, for example "Patient had *** abnormal laboratory tests (INR in Blood by Coagulation assay) in the last 12 months". Maybe a binary flag will be enough.

6. In addition, I think it would help to present this supplementary table 3 by groups: medicines, laboratory test, demographics, ICD10 Umbrella, CPT4 Code. It would be very interesting to see what features are relevant for the algorithm.

Outcome definition:

1. "Development of HF was defined as new assignment of an ICD-10 code for HF". Therefore, no ICD9 codes were considered for the definition of HF. am I right? I assumed all the systems were using ICD10 then. Supplementary table 1 includes some ICD9 codes, which was confusing.

2. As stated above, please clarify the training and test sets.

Model construction:

1. Hyper-parameter tuning: I haven't seen any reference to the hyper-parameters of the model. As per the API, there are many hyper-parameters that can be tuned: https://xgboost.readthedocs.io/en/latest/parameter.html. That is, a "development" set (in addition to the training and test set) should be put aside to tune these hyper-parameters. Alternatively, cross validation could be used in the training set using techniques such as grid search or random search to tune those hyperparameters.

a. Why did you decide to change the max_depth from the default value (6) to 5?

b. Why didn't you tune any of the others hyper-parameters or a subset of them? This could change the performance by much. For example, learning rate, number of estimators, type of regularization (L1 or L2), ...

2. Reproduction of the results: Which library, version and software did you use? Which type of machine?

Results:

1. In the methods you wrote: "Finally, we performed a univariate filtering step to eliminate features associated

78 with the outcome with a chi-squared test p-value >0.2." but in the results, it seems that the features were chosen by the XGBoost algorithm: "Of the 43,906 possible data features before feature reduction techniques were performed, the boost algorithm selected 339 for inclusion in the final model (Supplemental Table 3)." Please clarify.

2. I didn't understand this sentence: "The model also selected as weak classifiers features such as undergoing eye

surgery, laxative use, abnormal iron levels, and vitamin D use, to name a few."

Figure 2:

The confusion matrix was confusing. Please use standard names: Predicted versus True/

Reviewer #4: The authors report the development of risk prediction tool for development of HF in adults using a large state-wide CCHIE from the state of Maine. In terms of data, the author collects enough data for modeling and analysis; in terms of algorithm, the author uses the classical machine learning algorithm xgboost to model the probability of incident-HF in year two from data collected in year one, and then prospectively validated in year three. Here are some questions that may need to be explained:

1.Throughout the data set, the positive data (disease +) is much less than the negative data (disease -), only about 1%, the data is highly uneven. For this kind of binary classification problem of data imbalance, we should first down-sample the (disease -) data to keep the two types of data as balanced as possible before modeling. Because of the data imbalance, the Sensitivity and PPV shown in the confusion matrix of Figure2 are not high enough. How does the author think about this problem?

2.On page 6 start from Line 89, The authors use XGboost algorithm to build two models, but the reason for choosing this algorithm needs to be supplemented, why there is no other algorithm, such as SVM or fully connected neural network and so on.

3.Among the excluded patients, only the diagnosis or data of the previous year were excluded. If the patients did not come to see a doctor because of HF in the previous year, but had a previous history of HF (such as diagnosed earlier), how to exclude them? Will them be mistakenly included?

4.The table1 baseline data are too simple, so we should recompare the baseline data between the training group and the validation group, and whether there is any difference between the heart failure group and the non-heart failure group combined with the most important feature found by table2.

Reviewer #5: The manuscript describes an original application of machine learning for new-onset heart failure prediction in a 1-year timeframe on a large cohort of subjects.

Based on the manuscript in its current form, I do not have sufficient elements to know whether statistical analysis (especially machine learning) has been performed appropriately. The level of detail in which the methods and experimental procedures are described should be increased.

Most importantly, please add more details related to the machine learning experiments. For example:

- Is data reduction applied before the classification? If so, was it done only on the training set to avoid information leakage?

- What kind of cross-validation was used for model development?

- How was the final model derived?

- How were the XGBoost parameters chosen?

- The data set classes appear to be highly imbalanced: was this taken into account in model development?

Please provide further details about the "data reduction techniques to reduce dimensionality" mentioned in the Methods section.

The initial number of features is reported only in the Results section: I suggest mentioning it also in the Methods section, "Feature selection and preprocessing", together with the number of features resulting after dimensionality reduction and the univariate filtering step.

Please specify what criterion was followed to aggregate lab data into abnormal/normal.

When the AUC values are first reported, e.g. "0.797 [0.790-0.803]" etc, the "95% CI" statement inside the brackets is missing. Moreover, are the results reported as mean or median plus CI?

Minors:

- Please check the manuscript for proper spacing between words and references, spelling ("others models", "this algorithm could integrate into the HIE", etc.), and punctuation.

7. PLOS authors have the option to publish the peer review history of their article (what does this mean?). If published, this will include your full peer review and any attached files.

Reviewer #2: No

Reviewer #3: **Yes: **Oscar Perez-Concha

Reviewer #4: No

Reviewer #5: No

---

## [Author Response · Author response to Decision Letter 1]

3 Oct 2021

PONE-D-21-10920R1 REBUTTAL LETTER

Title “A prospectively validated novel risk prediction model for new onset heart failure utilizing a large statewide health information exchange”

RESPONSE: Title has been revised as “Case finding for patients at risk of new onset heart failure: utilizing a large statewide health information exchange EHR data to train and validate a risk prediction model”. 

RESPONSE: we logged on to https://www.editorialmanager.com/pone/ and select the 'Submissions Needing Revision' folder to include the following items when submitting our revised manuscript:

• A rebuttal letter that responds to each point raised by the academic editor and reviewer(s). This letter was uploaded as a separate file labeled 'Response to Reviewers'.

• A marked-up copy of your manuscript that highlights changes made to the original version. A separate file labeled 'Revised Manuscript with Track Changes' was uploaded.

• An unmarked version of your revised paper without tracked changes. A separate file labeled 'Manuscript' without track was uploaded.

RESPONSE to REVIEWER 2:

• The manuscript itself lacks a lot of literature review on related works.

RESPONSE: Additional literature of related works has been added to the METHODS per suggestion

• The authors should compare the performance results to previous studies on the same dataset.

RESPONSE: Our HIE dataset has not been analyzed by other groups yet, therefore, it would be technically challenging to compare on the same HIE dataset.

• The authors should propose more feature selection techniques to find out the optimal ones.

How did the authors perform hyperparameter optimization of the models?

RESPONSE: A new method section “Model revisited” was added to address these points.

• Machine learning-based model (i.e., XGBoost) has been used in previously biomedical studies such as PMID: 31987913 and PMID: 32942564. Thus, the authors are suggested to refer to more works in this description to attract a broader readership.

RESPONSE: The method section was revised to reference PMID: 31987913 and PMID: 32942564.

• There must have space before reference number.

RESPONSE: The text was revised to remove space before the reference number.

RESPONSE to REVIEWER 3:

I would invite the authors to tackle this problem by using some of the machine learning techniques that deal with imbalanced datasets, for example, using a Class Weighted XGBoost or Cost-Sensitive XGBoost.

Finally, the authors have seemed to just "plug" the XGBoost to the data without tuning any hyper-parameter. I would also invite the authors to re-visit this and tune some of the most important hyper-parameters of the XGBoost. This could significally improve the performance of the predictive algorithm.

RESPONSE: A new section under “Exploratory Model Analyses” was added to address these points.

Please find some other suggestions below:

Abstract:

1. Methods and Results: "A tree-boosting algorithm was developed": A tree-boosting algorithm was trained, rather than developed.

RESPONSE: revision was made throughout the main text.

Methods

Database and subject selection criteria:

1. I would use the terms "training and test" sets for the sets that you use for training and testing. I am not 100% clear which part of the data you use for training and which one for testing. Explain this in detail with dates and number of records. Table 1 will be ideal for that.

Every machine learning algorithm is tested in some data that the model hasn't seen during the training, but I wouldn't call it "prospectively validated" unless you collect the data prospectively, which as far as I understood, it is not the case here. This paper may be of interest: https://jamanetwork.com/journals/jamanetworkopen/fullarticle/2760438 (Prospective and External Evaluation of a Machine Learning Model to Predict In-Hospital Mortality of Adults at Time of Admission Nathan Brajer). They used the terms training and test sets to build the model, and then, they prospectively validated by validation the model with real-time data: "The model was integrated into the production electronic health record system and prospectively validated on a cohort of 5273 hospitalizations representing 4525 unique adult patients admitted to hospital A between February 14, 2019, and April 15, 2019." Therefore, in my opinion, you use the data as if it was collected prospectively, which is exactly the idea behind any test set, but you didn't integrate the model into production and validate prospectively.

I would remove the term "prospectively validated" from the manuscript and replace it with "test the algorithm" or validate it in the test set.

2. "Subjects in the model building/training cohort were randomly split into a 2/3 training and 1/3 retrospective prediction group, which was used to train separate models under differing feature sets (see below)." I was confused with the definition of the training and test set. In addition ,Table 1 doesn't specify the dates or the amount of patients (2/3 and 1/3) that falls in each group.

Table 1 says that the training cohort contains 497,470 patients, but Figure 1 says the 497,470 were divided into "observation period" (usually this is the training set) and prediction period (this is NOT usually part of the training set).

Finally, the term "validation cohort" is a bit confusing too, since "validation or development set" is commonly used for hyperparameter tuning. As stated above, I would change it to training and test. If validation cohort is preferred, please state clearly, with dates and number of patients, what sets you used for training, test and external testing or "validation". But I don't think it is either external (since the data comes from the same system) or prospective (as it is not collected prospectively).

I would remove the term "prospectively validated" from the manuscript and replace it with "test the algorithm" or validate it in the test set

2. "Subjects in the model building/training cohort were randomly split into a 2/3 training and 1/3 retrospective prediction group, which was used to train separate models under differing feature sets (see below)." I was confused with the definition of the training and test set. In addition ,Table 1 doesn't specify the dates or the amount of patients (2/3 and 1/3) that falls in each group.

RESPONSE: 

All the above points are related to the clarification of discovery and validation analytics. Therefore, we address collectively here. We summarize the reviewer’s input into two perspectives: (1) confusion as to the meaning of “prospective”; (2) need additional clarification for the retrospective (training) and prospective (validation) cohorts (Table 1). 

We agree with the reviewer that the manuscript needs additional clarification—overall we meant prospective to mean the algorithim was validated on data in a subsequent time period, but acknowledge that this is an unclear and potentially inaccurate definition. We therefore have removed all references to prospective validation and instead simply refer to this process as “validation”.

1. Title “A prospectively validated novel risk prediction model for new onset heart failure utilizing a large statewide health information exchange” was revised as “Identification of patients at risk of new onset heart failure: utilizing a large statewide health information exchange to train and validate a risk prediction model”.

2. In the method section, we used the terms of “discovery cohort” (i.e the old retrospective) and “validation cohort” (i.e the old prospective) to clarify. The discovery cohort was randomly partitioned into 1/3 for training and cross validation, 1/3 for calibration, and 1/3 for blind testing for performance evaluation. The final model performance was then measured on the subsequent year of data: the “validation” cohort.

3. We added an “Exploratory analysis” section to the methods and results. In these analyses we incorporated several reviewer suggestison on model development and tuning and reported the improved results.

4. Table 1 was updated in the main text to include more subject information.

1. Did you convert the ICD9 to ICD10 codes? You talked about using 5 or 3 digits with the ICD10. How do you deal with the ICD9 codes?

RESPONSE: Method section was revised to clarify that we only used ICD9 codes to exclude patients with historical record of having heart failure. 

2. "Finally, we performed a univariate filtering step to eliminate features associated

78 with the outcome with a chi-squared test p-value >0.2." You could use XGBoost for both purposes, 1) feature/dimensionality reduction (https://machinelearningmastery.com/feature-importance-and-feature-selection-with-xgboost-in-python/) and 2) Final model. It would be interesting to see which features the XGBoost presents as the most relevant.

RESPONSE: We incorporated this suggestion and reported this under additional exploratory analyses. The method text was revised to add a new section and the results were summarized in the limitation section and the supplementary materials. A table in the supplemental file (S3) is provided with all features selected by their data type and relative importance rank as requested. 

3. Did not you include patient characteristics such as gender? I think males tend to have higher incidence of HF.

RESPONSE: Gender is one of the discriminative features (Table 2, importance rank 17) used by XGboost in the trained model. 

4. "Medications were mapped to medication class using the Established Pharmacologic Class coding system". Were medications treated as binary 1-Medication was taken, 0 -Wasn't used, or did you use the amount of units, tablets,...?

In supplementary Table 3, the medications contain this text: Patient had *** medications , which makes me think you consider a continuous number. Please clarify.

RESPONSE: Medications were treated as features with binary values. Supplementary Table 3 was revised for clarity.

5. Exactly the same for lab test, for example "Patient had *** abnormal laboratory tests (INR in Blood by Coagulation assay) in the last 12 months". Maybe a binary flag will be enough.

RESPONSE: Laboratory tests were treated as features with binary values. Supplementary Table 3 was revised for clarity.

6. In addition, I think it would help to present this supplementary table 3 by groups: medicines, laboratory test, demographics, ICD10 Umbrella, CPT4 Code. It would be very interesting to see what features are relevant for the algorithm.

RESPONSE: Supplementary table 3 was revised as suggested.

Outcome definition:

1. "Development of HF was defined as new assignment of an ICD-10 code for HF". Therefore, no ICD9 codes were considered for the definition of HF. am I right? I assumed all the systems were using ICD10 then. Supplementary table 1 includes some ICD9 codes, which was confusing.

RESPONSE: Only ICD-10 was used in the modeling process. However, when enrolling the patients, we excluded patients with historical HF events encoded by ICD-9. We used ICD-9 codes under this context.

2. As stated above, please clarify the training and test sets.

Model construction:

1. Hyper-parameter tuning: I haven't seen any reference to the hyper-parameters of the model. As per the API, there are many hyper-parameters that can be tuned: https://xgboost.readthedocs.io/en/latest/parameter.html. That is, a "development" set (in addition to the training and test set) should be put aside to tune these hyper-parameters. Alternatively, cross validation could be used in the training set using techniques such as grid search or random search to tune those hyperparameters.

a. Why did you decide to change the max_depth from the default value (6) to 5?

b. Why didn't you tune any of the others hyper-parameters or a subset of them? This could change the performance by much. For example, learning rate, number of estimators, type of regularization (L1 or L2), ...

RESPONSE: We have expanded the methods section to include the hyper-parameter tuning process using a grid search method and the parameters chosen under “model construction and Tuning”

2. Reproduction of the results: Which library, version and software did you use? Which type of machine?

RESPONSE: Method section was revised to include a new sub section to address these points.

Results:

1. In the methods you wrote: "Finally, we performed a univariate filtering step to eliminate features associated 78 with the outcome with a chi-squared test p-value >0.2." but in the results, it seems that the features were chosen by the XGBoost algorithm: "Of the 43,906 possible data features before feature reduction techniques were performed, the boost algorithm selected 339 for inclusion in the final model (Supplemental Table 3)." Please clarify.

RESPONSE: The method section was revised to clarify that the first step was to remove features with a univariate chi-squared test, and the second step was to allow the XGBoost algorithim to select important features with an importance gain greater than 0.

2. I didn't understand this sentence: "The model also selected as weak classifiers features such as undergoing eye

surgery, laxative use, abnormal iron levels, and vitamin D use, to name a few."

RESPONSE: This sentence was removed.

Figure 2:

The confusion matrix was confusing. Please use standard names: Predicted versus True

RESPONSE: The figure was revised as suggested as suggested.

Reviewer #4: The authors report the development of risk prediction tool for development of HF in adults using a large state-wide CCHIE from the state of Maine. In terms of data, the author collects enough data for modeling and analysis; in terms of algorithm, the author uses the classical machine learning algorithm xgboost to model the probability of incident-HF in year two from data collected in year one, and then prospectively validated in year three. Here are some questions that may need to be explained:

1.Throughout the data set, the positive data (disease +) is much less than the negative data (disease -), only about 1%, the data is highly uneven. For this kind of binary classification problem of data imbalance, we should first down-sample the (disease -) data to keep the two types of data as balanced as possible before modeling. Because of the data imbalance, the Sensitivity and PPV shown in the confusion matrix of Figure2 are not high enough. How does the author think about this problem?

RESPONSE: The goal of our study was to create a clinically applicable model using real world EMR data to model the future clinical outcomes. We agree that it is common in data mining to have an imbalanced dataset. However, in our study we felt that the PPV and sensitivity were actually very good for a real-world test meant to identify potential cases of heart failure, not to definitively diagnose heart failure. The goal here is NOT for binary classification but to find patients at risk for targeted clinical outcomes. However, we agree that class imbalances do need to be accommodated in our models and therefore we included additional analysis to do so under “Exploratory analyses”. The results were summarized in the supplementary section. Indeed, we observed the modest improvement (p value, 0.01) of the overall predictive performance (ROC AUC), though we note that the PPV was unchanged and the specificity of the test was actually decreased. We elaborate in the limitations section that the choice of optimal test ultimately lies in the clinical use of the test, and that continued tuning/optimization of a prediction model to meet the need in question is required for implementation. 

2.On page 6 start from Line 89, The authors use XGboost algorithm to build two models, but the reason for choosing this algorithm needs to be supplemented, why there is no other algorithm, such as SVM or fully connected neural network and so on.

RESPONSE: We have tried different modeling algorithms (data not shown) including SVM and ultimately chose XGBoost for its modeling performance and computing efficiency.

3.Among the excluded patients, only the diagnosis or data of the previous year were excluded. If the patients did not come to see a doctor because of HF in the previous year, but had a previous history of HF (such as diagnosed earlier), how to exclude them? Will them be mistakenly included?

RESPONSE: In our study we excluded all patients with any prior history of HF in their record (even before one-year). However if a patient were new to the system and carried a diagnosis of HF from another system, than there is a potential for including them as at-risk for developing HF when in fact they already had that diagnosis. We have added this clarification to the limitation section.

4.The table 1 baseline data are too simple, so we should compare the baseline data between the training group and the validation group, and whether there is any difference between the heart failure group and the non-heart failure group combined with the most important feature found by table 2.

RESPONSE: Table 1 was revised as suggested.

Reviewer #5: The manuscript describes an original application of machine learning for new-onset heart failure prediction in a 1-year timeframe on a large cohort of subjects.

Based on the manuscript in its current form, I do not have sufficient elements to know whether statistical analysis (especially machine learning) has been performed appropriately. The level of detail in which the methods and experimental procedures are described should be increased.

Most importantly, please add more details related to the machine learning experiments. For example:

- Is data reduction applied before the classification? If so, was it done only on the training set to avoid information leakage?

RESPONSE:

We applied feature engineering pipeline to preprocess steps that transform raw data into features that can be used in machine learning algorithms on both training set and test set, and also perform feature selection based on transformed features in order to eliminate feature with low variation only for training set. And there should be no information leakage since no subject level data reduction was applied. For feature engineering, since raw data collection resulted in a massive number of potential coding features. Domain knowledge mapping was applied, medications were mapped to medication class using the Established Pharmacologic Class coding system; laboratory data were aggregated into “abnormal” and “normal” binary categories; and we performed an experiment to determine whether aggregating the 5-digit ICD-CM10-CM codes into the 3-digit code to the left of the decimal (henceforth known as the ICD-10 subheader code) would improve model performance and reduce dimensionality. Finally, we performed a univariate filtering step to eliminate features associated with the outcome with a chi-squared test p-value >0.2. 

We revised the method section to add a subsection under “Exploratory Analysis” to followed this and other reviewers’ input to fine tune the model and results were described in the Potential limitation section to compare the improvement and effectiveness.

- What kind of cross-validation was used for model development?

RESPONSE: 10-fold cross-validation was used when we used 1/3 of the discovery cohort to train the model. Relevant method section was revised.

- How was the final model derived?

RESPONSE: Method section was revised to clarify: “Subjects in the discovery cohort were randomly split into 1/3 training, 1/3 calibration, and 1/3 performance testing groups. The modeling, calibration and performance blind testing processes were used to minimize over-optimism of the test performance characteristics.”

- How were the XGBoost parameters chosen?

RESPONSE: We have clarified this in the methods section under “Model construction and tuning”: After hyper-parameter fine tuning process, the optimized hyper-parameter combination was found based on best model performance (the learning rate was set to 0.3, the depth of each tree was set to 5 and number of estimators was set to 500). Then, these optimized parameters were applied for XGBoost algorithm on entire training set to derive final model.

- The data set classes appear to be highly imbalanced: was this taken into account in model development?

RESPONSE: This was raised by another reviewer as well. We addressed this in exploratory analyses.

Please provide further details about the "data reduction techniques to reduce dimensionality" mentioned in the Methods section. The initial number of features is reported only in the Results section: I suggest mentioning it also in the Methods section, "Feature selection and preprocessing", together with the number of features resulting after dimensionality reduction and the univariate filtering step.

RESPONSE: The method section was revised to clarify as suggested in the “Feature standardization, reduction, and selection”.

Please specify what criterion was followed to aggregate lab data into abnormal/normal.

RESPONSE: Method section was revised to clarify this. “Laboratory data were provided from the HIE as “abnormal” and “normal” binary categories due data interoperability challenges requiring raw test values to be converted to binary abnormal/normal categorical variables via comparing test result value against the corresponding care providers’ test normal reference range.”

When the AUC values are first reported, e.g. "0.797 [0.790-0.803]" etc, the "95% CI" statement inside the brackets is missing. Moreover, are the results reported as mean or median plus CI?

RESPONSE: The text was revised as suggested. The results reported as median with 95% CI.

Minors:

- Please check the manuscript for proper spacing between words and references, spelling ("others models", "this algorithm could integrate into the HIE", etc.), and punctuation.

RESPONSE: The texts were revised accordingly.

7. PLOS authors have the option to publish the peer review history of their article (what does this mean?). If published, this will include your full peer review and any attached files.

Do you want your identity to be public for this peer review? For information about this choice, including consent withdrawal, please see our Privacy Policy.

Reviewer #2: No

Reviewer #3: Yes: Oscar Perez-Concha

Reviewer #4: No

Reviewer #5: No

---

## [Decision Letter · Decision Letter 2]

19 Nov 2021

Identification of patients at risk of new onset heart failure: utilizing a large statewide health information exchange to train and validate a risk prediction model

PONE-D-21-10920R2

Dear Dr. Ling,

We’re pleased to inform you that your manuscript has been judged scientifically suitable for publication and will be formally accepted for publication once it meets all outstanding technical requirements.

Kind regards,

Dylan A Mordaunt, MB ChB, FRACP, FAIDH

Academic Editor

PLOS ONE

Additional Editor Comments (optional):

Thank you for your resubmission. As you will have seen, we have had some change in reviewers, that often produces variability. I will address these as follows. The reviewers have provided some valuable feedback, and although my decision is to accept (which I detail below), I would suggest considering whether to include these in your final submission. Dr Perez-Concha has given a detailed discussion and it's a shame we don't have an editorial format that we could enable Dr Perez-Concha to expand on these, as I think they're valuable but shouldn't prevent publication under the PLoS One format.

With specific reference to PLoS One's criteria for publication (https://journals.plos.org/plosone/s/criteria-for-publication):

1. The study appears to present the results of original research.

2. Results appear not to have been published elsewhere.

3. Experiments, statistics, and other analyses are performed to a high technical standard and are described in sufficient detail. There are some additional comments from reviewers that do not represent critical flaws and are perhaps something to be addressed in post-publication review.

4. Conclusions are presented in an appropriate fashion and are supported by the data.

5. The article is presented in an intelligible fashion and is written in standard English.

6. The research meets all applicable standards for the ethics of experimentation and research integrity.

7. The article adheres to appropriate reporting guidelines and community standards for data availability.

Reviewers' comments:

Reviewer's Responses to Questions

**Comments to the Author**

1. If the authors have adequately addressed your comments raised in a previous round of review and you feel that this manuscript is now acceptable for publication, you may indicate that here to bypass the “Comments to the Author” section, enter your conflict of interest statement in the “Confidential to Editor” section, and submit your "Accept" recommendation.

Reviewer #2: (No Response)

Reviewer #3: (No Response)

Reviewer #4: (No Response)

Reviewer #5: All comments have been addressed

2. Is the manuscript technically sound, and do the data support the conclusions?

Reviewer #2: No

Reviewer #3: Partly

Reviewer #4: Partly

Reviewer #5: Yes

3. Has the statistical analysis been performed appropriately and rigorously? 

Reviewer #2: I Don't Know

Reviewer #3: I Don't Know

Reviewer #4: I Don't Know

Reviewer #5: Yes

4. Have the authors made all data underlying the findings in their manuscript fully available?

Reviewer #2: No

Reviewer #3: Yes

Reviewer #4: Yes

Reviewer #5: No

5. Is the manuscript presented in an intelligible fashion and written in standard English?

Reviewer #2: Yes

Reviewer #3: Yes

Reviewer #4: Yes

Reviewer #5: Yes

6. Review Comments to the Author

Reviewer #2: Some of my previous comments have been addressed. However, there are still some concerns as follows:

1. I previously asked for literature review in Introduction to show some previous works that focused on the same problem, not mean the related works in the Methods.

2. The authors should compare the performance results to previous studies on the same dataset. ==> If the authors aim to use their data and convince that their methods are good, they should replicate the other methods on their data to prove that. Currently there are some related works focusing on this prediction model, thus they must try and compare.

3. The authors should propose more feature selection techniques to find out the optimal ones.

Reviewer #3: Thank you for the opportunity to review this paper again. Thank you for having addressed my previous suggestions.

Abstract

•Conclusions: Instead of "passively" I would use the word routinely.

Methods

•Lines 74-77: “Laboratory data were provided from the HIE as “abnormal” and “normal” binary categories due data interoperability challenges requiring raw test values to be converted to binary abnormal/normal categorical variables via comparing test result value against the corresponding care providers’ test normal reference range”.

Questions:

a.Does this sentence mean that you do not know the criteria which were followed to aggregate lab data into abnormal/normal?

b.Did all the health providers (hospitals, outpatient clinics, …) follow the same criteria to convert numbers to normal and abnormal categories?

•The section Exploratory Model Analyses should be included in the section Model construction and tuning, as both sections deal with hyperparameters. Please create a table or summarize all the hyper-parameters that you have tuned, instead of finding this information scattered across the paper. I didn't have clear some of the steps that you followed and that's why I answered "I don't know" to the question of "3. Has the statistical analysis been performed appropriately and rigorously?"

•It will be very useful to include a “graph” or plot with the exact definition of “discovery” and “validation” cohort.

•Line 143: “the positive class, grid search a range of different class weightings (90, 95, 100, 110, 150)”. What was the value of the weight for the negative class? A value of 1? What is the meaning of a weight of 150?

Results

•Table 1. Reviewer 4, comment 4 suggested: “The table 1 baseline data are too simple, so we should compare the baseline data between the training group and the validation group, and whether there is any difference between the heart failure group and the non-heart failure group combined with the most important feature found by table 2”. You said you addressed this, but I don’t see that you listed the important features of table 2 within table 1. I agree I would add more features to the table and percentages to the numbers. You could add the top-ranked 25 known features associated with HF.

I think the main result of this study should be how difficult is to predict HF even with big amounts of data. I don’t think the main finding or result is a predictive algorithm per se. I would wonder how many clinicians would trust it. The sensitivity of 29.2 is a very low value. I don’t think AUC or specificity are very informative in this case, as the problem is highly imbalanced. I would suggest that the authors frame the question in terms of the methodology to predict HF, what we can be done with this method, and what we have to do in the future to predict HF more accurately.

For future work, it might be worth exploring the nature of the question, that is, prediction of HF in the next year. Maybe we need to understand better which predictors/features are going to predict HF more accurately, instead of feeding everything directly to the model. In addition, it could be beneficial to use several models and make a comparison.

Reviewer #4: Whether the model is used for diagnosis or screening, the nature of machine learning is the same. If the data is highly unbalanced, the model will learn more about the characteristics of the "majority" samples and ignore the "minority" samples. So I suggest adding an experiment to sample the "majority" samples so that the data are equalized and modeled again, and the PPV and sensitivity of your new model will be more valuable.

Reviewer #5: I would like to thank the authors for replying to my previous comments.

I suggest to rephrase the new paragraph “Laboratory data were provided from the HIE as “abnormal” and “normal” binary categories due data interoperability challenges requiring raw test values to be converted to binary abnormal/normal categorical variables via comparing test result value against the corresponding care providers’ test normal reference range.”, since it is hard to read.

Other newly added parts may also benefit from proofreading. I have no further issues.

7. PLOS authors have the option to publish the peer review history of their article (what does this mean?). If published, this will include your full peer review and any attached files.

Reviewer #2: **Yes: **Khanh N.Q. Le

Reviewer #3: **Yes: **Oscar Perez-Concha

Reviewer #4: No

Reviewer #5: No

---

## [Editor Report · Acceptance letter]

3 Dec 2021

PONE-D-21-10920R2 

Identification of patients at risk of new onset heart failure: utilizing a large statewide health information exchange to train and validate a risk prediction model 

Dear Dr. Ling:

I'm pleased to inform you that your manuscript has been deemed suitable for publication in PLOS ONE. Congratulations! Your manuscript is now with our production department. 

Kind regards, 

on behalf of

Dr. Dylan A Mordaunt 

Academic Editor

PLOS ONE